# Edge-wise Topological Divergence Gaps: Guiding Search in Combinatorial Optimization

## Abstract

We introduce a topological feedback mechanism for the Travelling Salesman Problem (TSP) by analyzing the divergence between a tour and the minimum spanning tree (MST). Our key contribution is a canonical decomposition theorem that expresses the tour-MST gap as edge-wise topology-divergence gaps from the RTD-Lite barcode. Based on this, we develop a topological guidance for 2-opt and 3-opt heuristics that increases their performance. We carry out experiments with fine-optimization of tours obtained from heatmap-based methods, TSPLIB, and random instances. Experiments demonstrate the topology-guided optimization results in better performance and faster convergence in many cases.

## 1 Introduction

The Travelling Salesman Problem (TSP) remains a cornerstone of combinatorial optimisation and a proving ground for machine-learning-augmented solvers. Despite impressive progress from neural construction policies (Vinyals et al., 2015; Kool et al., 2019) and reinforcement-learning (RL) refinements (da Costa et al., 2020; Chen et al., 2024), state-of-the-art pipelines still rely on *blind* post-hoc local search (e.g. 2-opt) to reach competitive tour lengths. These moves examine $O(n^2)$ edge pairs without principled guidance and dominate run-time on medium-size instances.

Building on recent advances in topological data analysis for graphs, in particular the *RTD-Lite* barcode (Tulchinskii et al., 2025), we prove a **canonical decomposition theorem**: the usual tour-MST length gap equals the sum of *non-negative, edge-wise topology-divergence gaps*. Each gap is the length of a bar in the RTD-Lite barcode and pinpoints how much a single tour edge deviates from its uniquely associated MST edge.

The Travelling Salesman Problem (TSP) is routinely framed as a *pure cost-minimisation task*: almost every heuristic, exact solver and learning approach measures progress solely by the total tour length. This perspective ignores *how* cities are connected as the tour evolves, even though decades of geometric and clustering work show that good tours first respect the instance's intrinsic connectivity before shaving residual distance (Zahn, 1971; Held & Karp, 1970). Consequently, contemporary neural solvers spend vast computation on blind local search to repair long-range "topological mistakes" that went undetected during learning (Kool et al., 2019; Fan et al., 2024).

**Topology as a missing optimisation signal.** We argue that explicit topological feedback can bridge this gap. Leveraging recent advances in graph topology, we employ the topology divergence framework (Tulchinskii et al., 2025) to compare a candidate tour with the instance's minimum-spanning tree (MST). Our principal theorem constructs a *canonical bijection* between tour edges and MST edges that decomposes the classical tour-MST length gap into *non-negative, edge-wise topological divergence gaps*. Each gap quantifies *the difference in scale at which MST clusters are connected by the tour*: large bars reveal tour edges that bridge distant vertex clusters much later than the MST does. By inspecting these bars we know, for every step of a constructive or improvement algorithm, which edge most distorts the natural connectivity structure.

**From length optimisation to topology-aware optimisation.** Building on this insight we develop two lightweight mechanisms: (i) a *topology-guided 2-opt* that cuts local-search swaps by always

attacking the highest divergence edge first, and (ii) a *gap-shaped Q-learning* framework whose reward combines tour length with the predicted topological gap, leading to faster convergence and better final tours. Crucially, both methods run in $O(n^2)$ extra time, matching the cost of distance-matrix construction.

Our experiments on TSPLIB and random Euclidean benchmarks demonstrate that adding this single topological signal delivers the first empirically validated link between persistent-homology theory and large-scale combinatorial optimization. We believe this topological perspective-and the practical gains it brings-opens a promising research direction beyond length-only objectives.

To the best of our knowledge this is the first work that (i) provides an *edge-wise* topological characterization of tour sub-optimality and (ii) demonstrates its utility in both classical and learning-based TSP solvers.

Our contributions are summarized as follows:

- **Canonical Tour-MST Decomposition**. We prove a theorem establishing a one-to-one mapping between tour edges and MST edges that decomposes the tour-MST length gap into a sum of non-negative, edge-specific divergence values (persistent bar lengths). This gives a principled, quantitative measure of how much each tour edge contributes to suboptimality.

- **Connection to LKH's $\alpha$-Score**. We show that the well-known $\alpha$-score used in the LKH-3 TSP solver is mathematically equivalent to a particular variant of the RTD-Lite barcode measure. This links our topological perspective to a successful traditional heuristic.

- **Topology-Guided Local Search**. We introduce topology-guided 2-opt and 3-opt heuristics that prioritize removing edges with large divergence gaps. On random Euclidean instances (up to 300 nodes) and TSPLIB benchmarks, this strategy consistently finds shorter tours and often converges in fewer iterations compared to standard 2-opt/3-opt.

- **Integration with Neural Solvers.** We demonstrate that topology-guided 2-opt can significantly improve the fine-optimization of tours generated by recent heatmap-based TSP solvers, even on very large instances (up to 10,000 nodes). Notably, our approach achieves better tours in less time than vanilla 2-opt post-processing on these neural outputs.

- **Topology-Shaped Reinforcement Learning**. We design an RTDL-gap-shaped reward for a Q-learning TSP solver, enabling an RL agent to receive per-edge feedback based on topological divergence. This reward shaping stabilizes training and reduces the final optimality gap by over 25% in fewer episodes compared to a purely length-based reward.

In summary, our work bridges topology and large-scale combinatorial optimization, providing a novel optimization signal that goes beyond traditional distance-based objectives. We believe this topological perspective opens a promising avenue for developing more structured and efficient search strategies in TSP and related problems.

## 2 RELATED WORK

**Classical TSP heuristics.** Local improvement schemes such as 2-opt, 3-opt, and Lin–Kernighan (LK) remain mainstays of high-quality TSP solvers (Croes, 1958; Lin & Kernighan, 1973; Helsgaun, 2000). LK and LKH rely on clever neighbourhood pruning to reduce the search space. The MST and 1-tree bounds have long provided global cost estimates (Held & Karp, 1970), yet have not been used to prioritize specific move choices. A comparison of first vs. best improvement in 2-opt algorithm was studied in Aloise et al. (2025); Hansen & Mladenović (2006).

**Neural and RL approaches.** Pointer networks (Vinyals et al., 2015), attention models (Kool et al., 2019), population RL (Team, 2023), and diffusion-based methods (Sanokowski et al., 2024) produce near-optimal tours but delegate final polishing to classical local search, which accounts for most wall-time (Fan et al., 2024). Attempts to learn local moves directly (da Costa et al., 2020) still lack a principled signal for *which* edge to change next.

**Topological data analysis (TDA) for graphs.** Persistence barcodes capture multiscale connectivity; RTD Barannikov et al. and its efficient variant RTD-Lite compare two weighted graphs in $O(n^2)$ time (Tulchinskii et al., 2025). Prior work used RTD as a scalar loss to preserve topology in neural

representations; our work is the first to exploit its *edge-level* information inside a combinatorial solver.

**Topology-aware optimisation.** Graph clustering via MST edge removal dates back to (Zahn, 1971). Recent "heat-map+MCTS" pipelines (Xia et al., 2024; Sun & Yang, 2023; Fu et al., 2021; Qiu et al., 2022; Min et al., 2023) acknowledge structural cues but still lack a rigorous measure of per-edge mis-alignment. We fill this gap by providing an exact, computable divergence for each edge and showing how it drives both local search and RL.

In summary, our paper bridges topological data analysis (TDA) and combinatorial optimization: it supplies a theoretically grounded, edge-wise error signal and demonstrates tangible performance gains in both classical and learning-based TSP solvers.

## 3 APPROACH

Consider an undirected weighted complete graph $G = (V, E)$ with weight $w(e)$ for edge $e$. Let $T_{\text{tour}}$ be a candidate TSP tour. For now, removing the longest tour edge $e_{max}$ consider $T_{\text{tour}}$ as a Hamiltonian path $T$ from a start city $s$ to an end city $t$, sometimes called an $(s, t)$-tour. Let $T_{\text{mst}}$ be a minimum spanning tree (MST) of $G$. We first formalize the relationship between $T_{\text{tour}}$ and $T_{\text{mst}}$ in terms of their edge sets and weights:

**Theorem 1.** *We construct a one-to-one correspondence $\phi$ between edges of $T$ and $T_{mst}$ so that the standard (TSP tour)-MST gap $L_{(s,t)-tour} - L_{mst}$ is decomposed as the natural sum of edge-wise non-negative topology divergence gaps:*

$$L_{(s,t)-tour} - L_{mst} = \sum_{e \in E(T_{mst})} w(\phi(e)) - w(e) \tag{1}$$

$$w(\phi(e)) - w(e) \geq 0, \text{ for any } e \in E(T_{mst}). \tag{2}$$

*In other words, every MST edge $e$ is paired with a unique tour edge $\phi(e)$ whose weight is greater than or equal to $w(e)$, and these weight differences exactly sum up to the gap between the $(s, t)$-tour length and the MST length.*

*Proof.* Assume for simplicity, that all edges in the union $E(T_{mst}) \cup E(T)$ have distinct weights, if some weights are the same then just choose an order on edges with the same weight. For $e \in E(T_{mst})$, let $A, B \in V$ denote the endpoints of $e$. Let $\tilde{e} \in E(T)$ denote the smallest weight edge of $T$ such that $A$ and $B$ are connected after addition of smaller or equal $w(\tilde{e})$ weight $T$−edges to $T_{mst}^{w<w(e)}$. In particular, $A$ and $B$ are connected by a path consisting of smaller than $w(e)$ weight $T_{mst}$−edges and bigger or equal $w(e)$ and smaller or equal $w(\tilde{e})$ weight $T$−edges. Notice that such path is not unique in general, but, by construction, it always contains the edge $\tilde{e}$. Then define $\phi(e) = \tilde{e}$. Conversely, given an edge $\tilde{e} \in E(T)$, let $\tilde{A}, \tilde{B}$, are its endpoints. Let $e' \in E(T_{mst})$ denotes the smallest weight edge of $T_{mst}$ such that $\tilde{A}, \tilde{B}$ are connected by a path after the addition of smaller or equal $w(e')$ weight $T_{mst}$−edges to $T^{w<w(\tilde{e})}$. Then the correspondence $\psi : \tilde{e} \mapsto e'$ is the inverse to $\phi$, see Appendix B. It follows that

$$\phi : E(T_{mst}) \to E(T), \tag{3}$$

is a one-to-one correspondence and that (2) holds. $\qquad \square$

It follows from the (Tulchinskii et al., 2025) definition of RTDL-Barcode of $(T_{\text{tour}}, G)$ that it is the sequence of intervals:

$$\text{RTDL-Barcode}(T_{\text{tour}}, G) = \{(w(e), w(\phi(e))) \mid e \in E(T_{mst})\}.$$

**Application to Hamiltonian cycles.** Let $T_{\text{tour}}$ be a Hamiltonian cycle. By definition, RTDL-Barcode$(A, B)$ is computed for two undirected weighted graphs $A, B$ with a bijection of vertices. The procedure for its computation (Algorithm 1, (Tulchinskii et al., 2025)), involves minimum spanning trees of $A, B$. To incorporate topological information into TSP solvers, we compute the RTDL-Barcode$(T_{\text{tour}}, G)$. The minimum spanning tree of $T_{\text{tour}}$ is $T_{\text{tour}} \setminus \{e_{max}\}$ where $e_{max}$ is an edge of $T_{tour}$ having the largest weight. The Theorem 1 implies a bijection $\psi$ between $E(T_{\text{tour}}) \setminus \{e_{max}\}$ and $E(T_{mst})$. For each $e \in E(T_{\text{tour}})$ we define the penalty

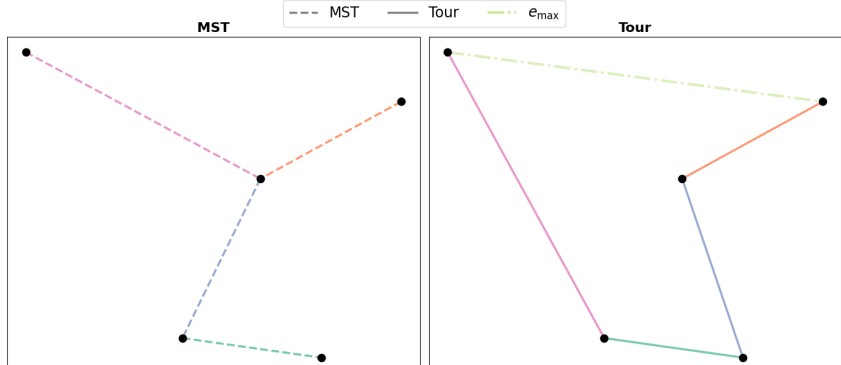

Figure 1: Visualization of the mapping between edges of the Hamiltonian cycle $T_{\text{tour}}$ and the minimum spanning tree $T_{\text{mst}}$. Each tour edge (except $e_{\max}$) is bijectively matched to an MST edge.

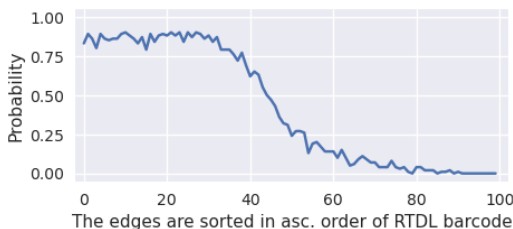

Figure 2: Probability of belonging to an optimal tour vs. RTDL barcode of an edge.

$p(e) = w(e) - w(\psi(e)) \geq 0$ as a measure of its "badness", that is, how more lengthy it is than the minimum spanning tree edge. A penalty for $e_{max}$ is not defined by this procedure. We set $p(e_{max}) = \min_{e \in \{x \in E(T_{tour}) \,|\, p(x) > 0\}} p(e)$. This value exhibited the best performance in computational experiments. A visualization of the described mapping between MST and tour edges is provided in Figure 1.

RELATION BETWEEN RTDL BARS AND $\alpha$-SCORE

$\alpha$-score is used for edge selection in the LKH-3 algorithm (Helsgaun, 2000) for TSP and VRP problems. $\alpha$-score is another measure of edge's "badness". Let $G$ be a weighted graph with weights $w_{i,j}$, $G_1 \subset G$ a subgraph without an arbitrary vertex "1" and incident edges. For an edge $(i,j)$:

$$\alpha(i,j) = L(R^+(i,j)) - L(R), \qquad (4)$$

where $R$ is the minimal 1-tree of $G$, and $R^+(i,j)$ is the the minimal 1-tree of $G$ with a restriction to contain the edge $(i,j)$, $L(\cdot)$ is a length function. We establish connection between $\alpha$-score and RTDL barcode in the following

**Proposition 1.** For $i, j \neq 1$, $\alpha(i,j)$ equals to the length of a bar in the RTDL-Barcode($G_1^{i,j}, G_1$), corresponding to an edge $(i,j)$. Both $G_1$, $G_1^{i,j}$ are graphs having all the vertices of $G$ without the vertex 1. $G_1$ is defined above, and $G_1^{i,j}$ has only one edge $(i,j)$ with a weight $w_{i,j}$.

The proof is provided in Appendix C. The main difference between our edge penalty $p(e)$ and the $\alpha$-score is that our penalty is tour-dependent. Also, contrary to the $\alpha$-score, our penalties decompose the standard tour length-MST gap into the sum of nonnegative gaps for the tour edges.

| Method | TSP-100 | | TSP-200 | | TSP-300 | |
|---|---|---|---|---|---|---|
| | Length | Time(s) | Length | Time(s) | Length | Time(s) |
| LK | 8.022 | 17.49 | 11.176 | 111.06 | 13.672 | 350.95 |
| 2-opt | 8.244 | 1.01 | 11.465 | 12.25 | 13.940 | 47.51 |
| 2-opt+RTDL | 8.226 | 2.84 | 11.356 | 18.08 | 13.798 | 62.30 |
| 2-opt+RTDL+opt(D) | **8.170** | 2.77 | **11.303** | 16.81 | **13.654** | 57.03 |
| 3-opt | 8.137 | 12.30 | 11.284 | 160.19 | 13.747 | 701.86 |
| 3-opt+RTDL | 8.082 | 13.49 | 11.157 | 131.50 | 13.594 | 545.50 |
| 3-opt+RTDL+opt(D) | **8.011** | 11.43 | **11.126** | 104.61 | **13.546** | 456.67 |

Table 1: 2D Euclidean TSP. Improvements of 2-opt and 3-opt with RTDL barcodes.

## 4 EXPERIMENTS

### 4.1 RTDL BARCODES AND OPTIMAL TOURS

How the value of an edge's RTDL barcode is related to the TSP solution? To study this dependency, we generated 100 random 2-dimensional TSP problems and found 1) non-optimal tours by running 2-opt procedure with varying number of iterations 2) optimal tours by running Concorde[1]. Then, for each non-optimal tour we ordered edges in an ascending order of their RTDL barcodes and estimated an empirical probability for an edge to belong to an optimal tour. Figure 2 shows that edges with a low barcode tend to belong to an optimal tour with high probability. See Appendix A for more results.

### 4.2 IMPROVEMENTS IN 2-OPT, 3-OPT

We apply RTDL-barcode to improve 2-opt and 3-opt algorithms. In this simplified setting we can test the influence of RTDL-derived weights of edges alone without interfering factors like complex training pipelines. The value of RTDL-barcode corresponding to a tour's edge defines its "badness". As a candidate for an optimal tour, the natural idea is to try to remove such edges from the tour before the others. In the vanilla 2-opt (Algorithm 1 in Appendix) the order of picking edges which Algorithm attempts to remove is not specified, typically it is sequential. We modify the 2-opt algorithm by picking edges in a descending order by their RTDL barcodes (Algorithm 2 in Appendix). The modified algorithm is denoted by 2-opt+RTDL.

**Euclidean TSP.** Table 1 presents experimental results for 2D Euclidean TSP (cities are sampled uniformly in a unit square). We report average tour length and average optimization time. All the results are averaged over 100 trials. For all the settings, an ordering by RTDL leads to a shorter tour with a cost of slightly longer time. For 3-opt algorithm, an ordering by RTDL both improves tour length and optimization time.

Also, we apply an heuristic from Helsgaun (2000). They noted that given scalars associated with vertices $\pi_i$ one can modify the distance matrix $d_{i,j} \leftarrow d_{i,j} + \pi_i + \pi_j$. It shifts different tours total lengths by the same constant. Thus, the optimal tour doesn't change. The $\pi$ values are optimized by a subgradient optimization procedure pushing the nodes of 1-tree to have degree 2, that is, more similar to a tour. Obviously, steps of standard 2-opt and 3-opt doesn't change after such modification. However, steps of 2-opt/3-opt+RTDL do change. In Table 1 this heuristic is denoted by *opt(D)* and it is shown to improve results even further.

We conclude that the ordering by RTDL barcodes always brings some improvement in objective without a significant increase of an execution time. Figure 3a shows that for 2-opt ordering by RTDL results in less number of trials per iteration – number of checked pairs of edges before tour improvement. For 3-opt, ordering by RTDL results in faster convergence, see Figure 3b. See ablation study in Appendix F.

**Non-metric TSP.** We also have carried out experiments with non-metric TSP (least weight Hamiltonian cycle problem), where edges' weights are sampled from $(0, 1)$. Again, an addition of RTDL always brings an improvement, see Table 5 in Appendix.

---

[1]https://www.math.uwaterloo.ca/tsp/concorde/

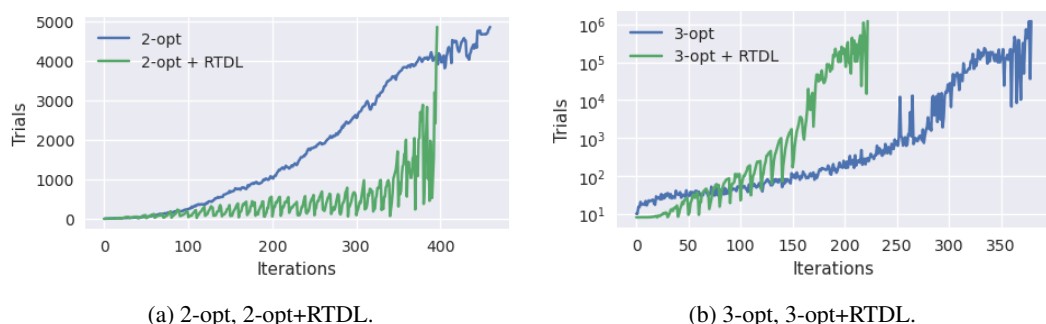

(a) 2-opt, 2-opt+RTDL.  (b) 3-opt, 3-opt+RTDL.

Figure 3: Avg. number of trials per iteration.

**TSPLib instances.** We also conducted experiments on tasks from TSPLib. For each task, optimization was run 100 times with different initial tours, see Table 3. For certain initial tours, the plain 2-opt algorithm struggled to reach the optimal solution and occasionally exceeded its iteration limit, whereas 2-opt+RTDL never encountered such issues under the same initial conditions. See Appendix D for more details.

## 4.3 RTDL FOR HEATMAP-GUIDED 2-OPT

In the case of large-scale TSP, there is a separate line of solvers that do not predict the optimal tour directly but rather generate a heatmap. Each entry $\Phi_{i,j}$ of the $n \times n$ heatmap $\Phi$ reflects the relevance of the edge $(i, j)$ to the optimal solution. In order to obtain a feasible solution given a heatmap, specialized decoding strategies are used (Sun & Yang, 2023). In this work, we follow Sun & Yang (2023) and further improve the initial tour obtained through greedy decoding of generated heatmaps with the 2-opt and 2-opt + RTDL algorithms. With this experiment, we aim to to verify whether RTDL-based edge traversal further improves 2-opt when initial tour is already a good approximation of the optimal solution.

Following the work Xia et al. (2024), we use the heatmaps learned by the DIFUSCO (Sun & Yang, 2023), ATT-GCN Fu et al. (2021), DIMES (Qiu et al., 2022), UTSP (Min et al., 2023), and SoftDist (Xia et al., 2024) approaches. We use all available problems for testing: 128 problems of size 500, 128 problems of size 1000 and 16 problems of size 10000. For each model and problem size, we report the average length of optimized tours and the average time of tour optimization. As shown in Table 2, in most cases, the proposed 2-opt + RTDL leads to solutions with a shorter average tour length and a more efficient computation. ==Heatmap-based methods are slightly worse than LKH-3 and Concorde which reflects the current performance of this family of methods.== Figure 4 further confirms that 2opt + RTDL provides faster convergence for all models. For an additional examination, we also provide the average length of the initial tours through heatmaps' greedy decoding in Table 6. For visual analysis, we provide the initial tours along with 2-opt and 2-opt + RTDL optimized tours for the same TSP-500 problem and all models in Figure 5. We find that RTDL updates every 5 (5, 100) iteration and $N = 500$ ($N = 1000$, $N = 10000$) for TSP-500 (TSP-1000, TSP-10000) works universally well for all models.

==The sensitivity analysis is presented in Figure 8. We can note that 2-opt+RTDL with Freq=1 (i.e., update on each iteration) typically leads to higher computation time, for all values of granularity. In the majority of cases, the average tour length for Freq=1 can be matched or even improved with Freq=5, 10 with appropriate granularity with sufficiently smaller computation time. Overall, for Freq¿1, more frequent updates lead to smaller average tour length. In general, the larger the granularity, the slower is computation, although this effect is less pronounced for higher frequencies. Up to some threshold frequency, larger granularity leads to higher average tour length. Above the threshold frequency, larger granularity leads to decreased average tour length. This threshold frequency depends on model and problem size.==

Table 2: Average length and running time of 2-opt and 2-opt + RTDL algorithms applied initial tours obtain through heatmap greedy decoding.

| Method | TSP-500 | | TSP-1000 | | TSP-10000 | |
|---|---|---|---|---|---|---|
| | Length | Time(s) | Length | Time(s) | Length | Time(s) |
| Baselines | | | | | | |
| Concorde | 16.55 | 18.73 | 23.12 | 201.01 | N/A | N/A |
| LKH-3 | 16.55 | 41.75 | 23.12 | 90.94 | 71.77 | 2698.38 |
| DIFUSCO | | | | | | |
| 2-opt | 16.98 | 0.38 | 24.01 | 5.78 | 75.87 | 7027.63 |
| 2-opt + RTDL | **16.88** | 0.23 | **23.60** | 1.76 | **74.28** | 865.5 |
| ATT-GCN | | | | | | |
| 2-opt | 17.63 | 1.09 | 24.72 | 12.54 | 77.36 | 17567.5 |
| 2-opt + RTDL | **17.3** | 0.55 | **24.2** | 3.05 | **75.54** | 1567.69 |
| DIMES | | | | | | |
| 2-opt | 17.74 | 1.66 | 24.84 | 16.05 | 77.68 | 25363.25 |
| 2-opt + RTDL | **17.35** | 0.73 | **24.31** | 3.56 | **75.3** | 1797 |
| UTSP | | | | | | |
| 2-opt | 17.54 | 0.93 | 24.57 | 10.20 | – | – |
| 2-opt + RTDL | **17.25** | 0.56 | **24.12** | 2.95 | – | – |
| SoftDist | | | | | | |
| 2-opt | 17.41 | 0.75 | 24.31 | 6.43 | 75.41 | 5264.88 |
| 2-opt + RTDL | **17.24** | 0.49 | **24.09** | 2.65 | **74.65** | 1549.5 |

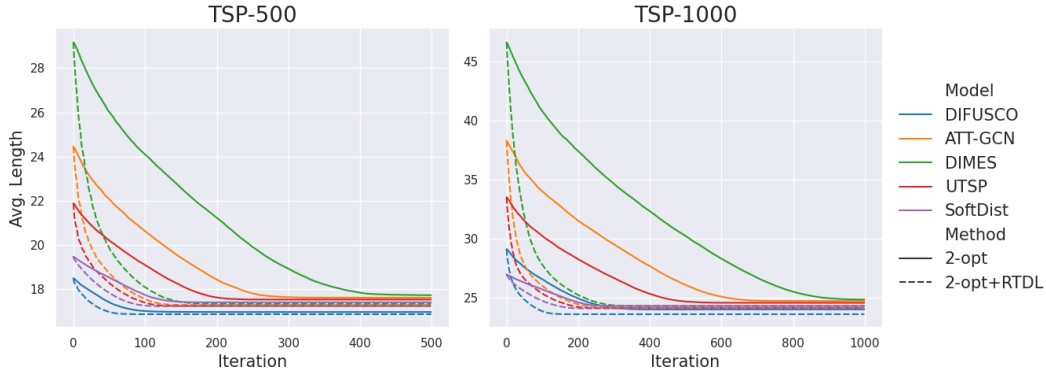

Figure 4: Evolution of average length during tour optimization for heatmap-guided 2-opt and 2opt + RTDL.

## 4.4 RTD FOR DEEP Q-LEARNING

To further evaluate the potential of incorporating topological insights into the Traveling Salesman Problem (TSP), we design a series of experiments based on a popular class of reinforcement learning (RL) methods-Deep Q-Networks (DQN). Our aim is to assess both a baseline learning strategy and a modified approach that takes advantage of additional topological information through reward tuning.

For these experiments, we randomly generated TSP tasks with different numbers of cities and then asked our models to iteratively construct tours multiple times for each instance independently. Each model is given 600 episodes to build a tour from scratch, and we record their best results.

In the DQN framework, the central idea is to learn a Q-function, $Q(s, a)$, which approximates the expected cumulative reward of taking action $a$ in state $s$ and then following the current policy thereafter. In the context of TSP, the "state" corresponds to a partially constructed tour, while the "action" represents choosing the next city to visit. By updating the Q-function through repeated interactions, the agent gradually learns to prefer actions that are more likely to lead to shorter tours, thereby

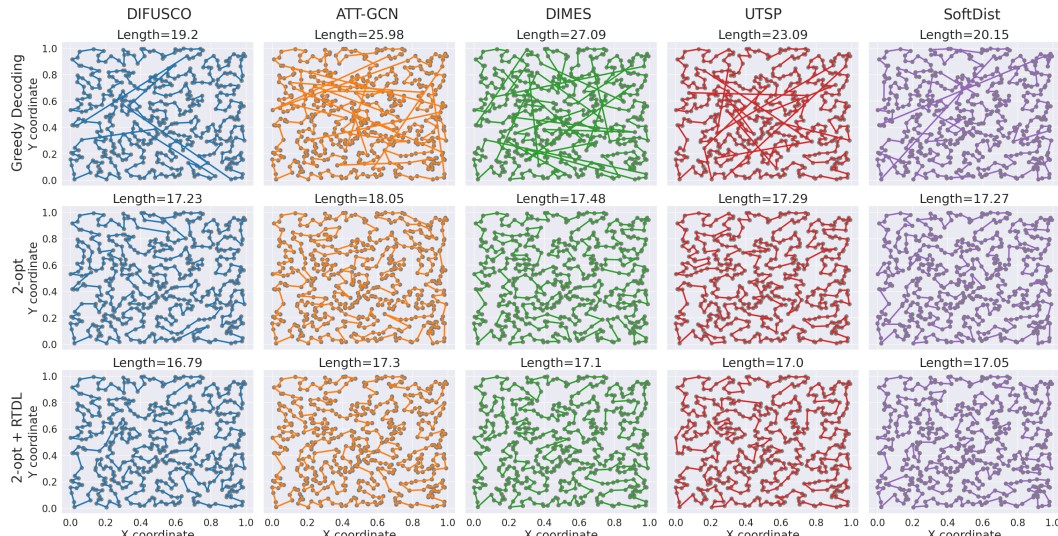

Figure 5: Example of solutions for the same TSP-500 problem through different models and optimization methods. For each model, we provide initial tour obtained through heatmap greedy decoding, 2-opt and 2-opt + RTDL optimized tours.

Table 3: Experimental results of 2-opt, 2-opt+RTDL, and 2-opt+RTDL-Full compared against Concorde on TSPlib tasks. All algorithms were run under the same time and iteration limits. We also report the GAP (%) relative to the best solution obtained by Concorde. Details of the algorithms are provided in Appendix C.

| | 2-opt | | | 2-opt+RTDL-Full | | | 2-opt+RTDL | | |
|---|---|---|---|---|---|---|---|---|---|
| Name | Tour len. | Time | GAP (%) | Tour len. | Time | GAP (%) | Tour len. | Time | GAP (%) |
| ulysses16 | $74.54 \pm 0.48$ | 0.02s | 0.6 | $74.51 \pm 0.41$ | 0.06s | 0.54 | $\mathbf{74.49 \pm 0.46}$ | 0.06s | **0.52** |
| ulysses22 | $76.28 \pm 0.43$ | 0.05s | 0.8 | $\mathbf{76.18 \pm 0.37}$ | 0.11s | **0.64** | $76.25 \pm 0.51$ | 0.13s | 0.78 |
| att48 | $34842.07 \pm 648.93$ | 0.20s | 3.9 | $\mathbf{34632.78 \pm 608.78}$ | 0.79s | **3.3** | $34848.44 \pm 609.44$ | 1s | 4.0 |
| eil51 | $451.66 \pm 31.26$ | 0.23s | 4.9 | $450.84 \pm 7.54$ | 0.44s | 4.9 | $\mathbf{446.68 \pm 7.82}$ | 0.58s | **3.9** |
| berlin52 | $8150.23 \pm 240.14$ | 0.27s | 8.0 | $8133.56 \pm 217.95$ | 0.53s | 7.0 | $\mathbf{7957.48 \pm 237.26}$ | 0.69s | **5.5** |
| st70 | $714.90 \pm 46.18$ | 0.63s | 5.5 | $\mathbf{705.53 \pm 10.45}$ | 1.02s | **4.1** | $709.45 \pm 11.09$ | 0.62s | 4.7 |
| eil76 | $604.50 \pm 37.62$ | 1.12s | 11.0 | $\mathbf{583.92 \pm 11.66}$ | 1.01s | **7.2** | $586.20 \pm 11.36$ | 1.83s | 7.6 |
| pr76 | $113227.56 \pm 12746.41$ | 0.81s | 4.7 | $113392.47 \pm 2581.35$ | 1.1s | 4.8 | $\mathbf{112657.0 \pm 2221.49}$ | 1.56s | **4.2** |
| gr96 | $541.12 \pm 12.91$ | 1.56s | 5.6 | $543.32 \pm 12.71$ | 1.66s | 6.1 | $\mathbf{538.54 \pm 12.01}$ | 2.6s | **5.1** |
| kroC100 | $22009.93 \pm 553.36$ | 1.95s | 6.1 | $22020.69 \pm 528.93$ | 1.7s | 6.1 | $\mathbf{21815.72 \pm 598.33}$ | 2.79s | **5.1** |

improving its decision-making process across episodes. Details of the algorithm are provided in Appendix E.

In our setup, the reward is dense: at each step the agent receives a negative edge weight corresponding to the cost of the chosen edge. This ensures that the cumulative reward over an episode is exactly the negative tour length. To incorporate topological information, we compute the RTDL-Barcode$(T_{\text{tour}}, G)$, where $T_{\text{tour}}$ is the constructed Hamiltonian cycle and $G$ is the full graph of cities. Following Theorem 1, each tour edge $e \in E(T_{\text{tour}}) \setminus \{e_{\max}\}$ is paired with an MST edge via the bijection $\psi$, while the pair for $e_{\max}$ is defined independently. This allows us to assign a per-edge topological reward

$$r_{\text{RTDL}}(e) = w(e) - w(\psi(e)) \geq 0.$$

This value is added step-wise to the environment reward at the moment the edge is selected. These penalties are then distributed as additional per-step rewards, effectively guiding the agent towards topologically meaningful tours while preserving the standard length-based reward.

To ensure fairness, all models are trained on identical city configurations with fixed starting and ending points, and are evaluated under equivalent conditions. Additionally, we benchmark the learned solutions against the global optimum computed using the Concorde TSP solver. The experiments, summarized in Figure 6, demonstrate that RTDL-based reward shaping significantly stabilizes and improves the training of DQN agents. The baseline DQN quickly plateaus and even diverges after

about 300 episodes, whereas DQN+RTDL remains stable and continues to reduce the tour length. Overall, DQN+RTDL consistently produces shorter tours and exhibits lower variance across seeds compared to the baseline. The numerical results in Table 4 corroborate these findings across different TSP sizes.

Table 4: Mean tour lengths for different TSP sizes comparing the baseline DQN and the RTDL-enhanced variant.

| Method | TSP-50 | TSP-70 | TSP-100 |
|---|---|---|---|
| DQN (base) | 8.76 | 10.31 | 18.66 |
| DQN + RTDL | **7.65** | **8.46** | **11.75** |

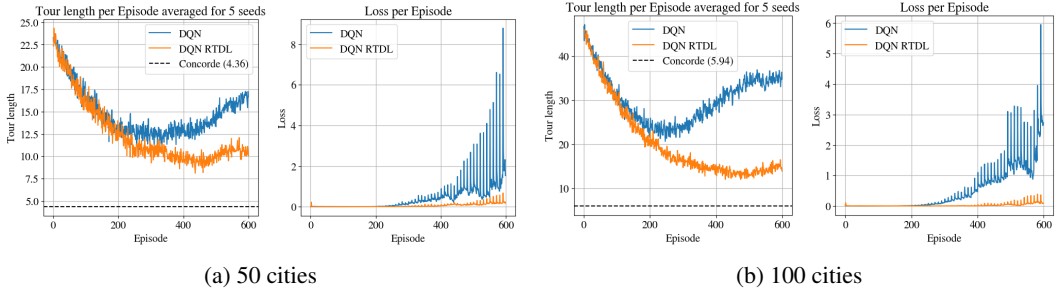

(a) 50 cities                    (b) 100 cities

Figure 6: Learning curves of the baseline DQN and DQN+RTDL on TSP instances with 50 and 100 cities. Each curve is averaged over five random initializations.

## 5 CONCLUSION

In this work, we have introduced a novel topological perspective on the TSP problem by decomposing the classical tour-MST total gap into edge-wise divergence contributions. We successfully used topological signals to construct topology-guided 2-opt, 3-opt leading to lower tour lengths for random Euclidean instances and TSPLib benchmark. Moreover, topology-guided 2-opt provided improvements for fine-optimization of tours generated by TSP solvers based on heatmaps, up to 10000 nodes. Using this insight, we proposed a topology-aware reinforcement learning framework in which each MST edge receives a localized reward proportional to its divergence contribution. Empirically, this fine-grained reward allocation accelerates convergence, yields more stable learning curves, and ultimately improves tour quality compared to conventional, monolithic reward schemes. Furthermore, we established an equivalence between the $\alpha$-score from the LKH-3 algorithm and a specific variant of the RTDL barcode, strengthening the link between our topological methods and established heuristic techniques. By bridging persistent homology with modern RL, our work opens a new avenue for incorporating global graph structure into deep reinforcement learning for combinatorial optimization.

## 6 REPRODUCIBILITY STATEMENT

The source code for the experiments is available in the supplementary material. Hyperparameters are either disclosed in Section 4, or were equal to defaults in code.

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

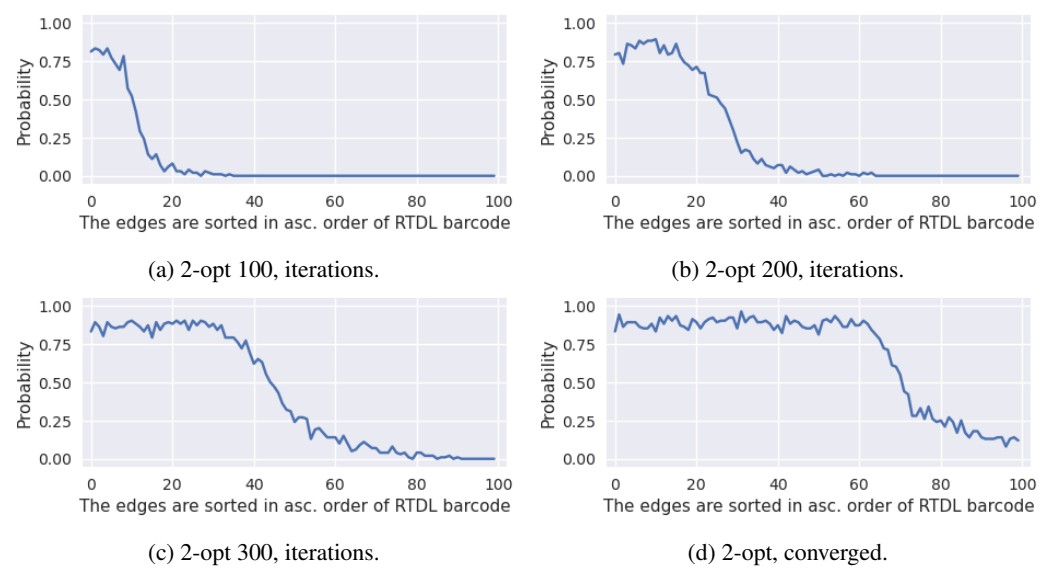

(a) 2-opt 100, iterations.

(b) 2-opt 200, iterations.

(c) 2-opt 300, iterations.

(d) 2-opt, converged.

Figure 7: Probability of belonging to an optimal tour vs. RTDL barcode of an edge.

## A    ADDITIONAL RESULTS

Figure 7 shows probabilities for an edge to belong to an optimal tour, edges are taken from 2-opt optimization with varying number of iterations. Table 5 shows results for of 2-opt and 3-opt for non-metric TSP problems.

## B    DETAILS OF THEOREM 1 PROOF

Denote by $\gamma^e$ the path that connects the edge $e \in E(T_{mst})$ endpoints $A$ and $B$ and consist of smaller than $w(e)$ weight $T_{mst}-$edges and bigger or equal $w(e)$ and smaller or equal $w(\tilde{e})$ weight $T-$edges, with the smallest weight $w(\tilde{e})$. Denote by $\tilde{\gamma}^{\tilde{e}}$ the path that connects the edge $\tilde{e} \in T$ endpoints $\tilde{A}$ and $\tilde{B}$ and consist of smaller or equal $w(e')$ weight $T_{mst}-$edges and smaller than $w(\tilde{e})$ weight $T-$edges, with the smallest weight $w(e')$. If for such a path $w(e') < w(e)$ then the union $\tilde{\gamma}^{\tilde{e}} \cup \gamma^e \setminus \tilde{e}$ would be a path connecting the vertices $A, B$ while consisting of smaller than $w(e)$ weight $T_{mst}-$edges and bigger or equal $w(e)$ and strictly *smaller* than $w(\tilde{e})$ weight $T-$edges, which contradicts the definition of $\tilde{e}$. Hence the path $e \cup \gamma^e \setminus \tilde{e}$ satisfies the conditions for path $\tilde{\gamma}^{\tilde{e}}$ with $w(e') = w(e)$, and $e'$ coinciding with $e$. Hence $\psi(\phi(e)) = e$, and the maps $\phi, \psi$ are bijective.

## C    RELATION OF RTDL BARS AND ALPHA-SCORE

Let $G$ be a weighted graph with weights $w_{i,j}$. Consider a subgraph $G_1 \subset G$ without an arbitrary vertex "1" and incident edges.

**Definition**. The 1-tree is a spanning tree of $G_1$ with the vertex 1 and two edges connecting the vertex 1 and $G_1$.

$\alpha$-score is used for edge selection in the state-of-the-art LKH algorithm Helsgaun (2000) for TSP and VRP problems. $\alpha$-score is a measure of edge's "badness". Edges with low $\alpha$-score tend to belong to the optimal tour with high probability. For an edge $(i, j)$:

$$\alpha(i, j) = L(R^+(i, j)) - L(R),$$

where $R$ is the minimal 1-tree of $G$, and $R^+(i, j)$ is the the minimal 1-tree of $G$ with a restriction to contain the edge $(i, j)$, $L(\cdot)$ is a length function.

**Proposition.** For $i, j \neq 1$, $\alpha(i,j)$ equals to a length of a bar in the RTDL-Barcode$(G_1^{i,j}, G_1)$, corresponding to an edge $(i,j)$. Both $G_1$, $G_1^{i,j}$ are graphs having all the vertices of $G$ without the vertex 1. $G_1$ is defined above, and $G_1^{i,j}$ has only one edge $(i,j)$ with a weight $w_{i,j}$.

*Proof.* An algorithm for calculating RTDL-Barcode(A, B) is the following (Tulchinskii et al., 2025). A precondition of RTDL evaluation is a bijection between vertices of graphs $A, B$. The auxiliary graph $C$ containing a union of edges of $A, B$ and weights on edges $w_e^C = min(w_e^A, w_e^B)$ is constructed. When an edge is missing, its weight is considered $+\infty$. Then, the Kruskal's algorithm for finding the MST of the graph $C$ is executed. Let $e^C$ be an edge from the MST of the graph $C$. The edge $e^C$ connects two connected components $C_1, C_2$. In the Algorithm of RTDL calculation, edges from the MST of $A$ are added in an increasing order by a weight. After adding some edge $e^A$ a path connecting $C_1$ and $C_2$ (any of their vertices) appears. Thus, $e^C$ is paired with $e^A$. Together they form a bar $(e^C, e^A)$ in the RTDL-Barcode. Note, that always $w_e^C \leq w_e^A$. If $e^A \neq e^C$, then after adding $e^A$ to $C_1 \cup C_2$ a cycle appears. As shown in (Helsgaun, 2000), the 1-tree $R^+(i,j)$ (if $i, j > 1$) can be obtained from the $R \cup e^A$ by removing the longest edge in the aforementioned cycle. Obviously, such edge is $e^C$. Thus, $\alpha(i,j) = w_e^A - w_e^C$. □

In Table 3 and Appendix D, we also evaluate two variants of 2-opt+RTDL:

- **2-opt+RTDL-Full** - described in Algorithm 2, but applied to all edges during each optimization stage.
- **2-opt+RTDL** - described in Algorithm 2, where edges are selected in batches of size 10 for each optimization cycle.

---

**Algorithm 1:** 2-opt algorithm

Initialize `tour`;

**repeat**
  START:
  improved ← false;
  **foreach** $e_1$ *in* `tour` **do**
    **foreach** $e_2$ *in* `tour` **do**
      tour_new ← tour;
      remove $e_1, e_2$ from `tour_new` and rewire it;
      **if** *Len(tour_new) < Len(tour)* **then**
        tour ← tour_new;
        improved ← true;
        goto START:
      **end**
    **end**
  **end**
**until** *improved*;

**return** `tour`

---

## D   ROBUSTNESS ANALYSIS OF 2-OPT AND RTDL VARIANTS

In this section, we conduct extensive experiments with 2-opt, 2-opt+RTDL, and 2-opt+RTDL-Full to evaluate their ability to converge from different initial tours.

We observe that plain 2-opt sometimes fails to converge to the global optimum from certain initial tours, whereas 2-opt+RTDL consistently avoids such cases. Figure 9 shows the distributions of convergence times and resulting tour lengths for the 76-city Christofides/Eilon instance, while Table 7 summarizes the frequency of extreme cases across a broader set of problems.

In particular, we report two types of failures: (i) runs where the tour length exceeds 10% above the global optimum ("GAP > 10%"), and (ii) runs that exceed the time budget of 20 seconds ("Time

---

**Algorithm 2:** 2-opt+RTDL algorithm

---

Initialize `tour`;
$N \leftarrow \min(10, |\texttt{tour}|)$
**repeat**
    START:
    improved $\leftarrow$ false ;
    $E \leftarrow$ edges from `tour`[0:N] sorted by RTDL barcodes in desc. order;
    **foreach** $e_1$ *in E* **do**
        **foreach** $e_2$ *in E* **do**
            `tour_new` $\leftarrow$ `tour`;
            remove $e_1, e_2$ from `tour_new` and rewire it;
            **if** *Len(`tour_new`) < Len(`tour`)* **then**
                `tour` $\leftarrow$ `tour_new` ;
                improved $\leftarrow$ true ;
                goto START:
            **end**
        **end**
    **end**
    **if** $N < |tour|$ **then**
        $N \leftarrow \min(N + 10, |\texttt{tour}|)$;
        improved $\leftarrow$ true ;
    **end**
**until** *improved*;
**return** `tour`

---

| Method | TSP-100 | | TSP-200 | | TSP-300 | |
|---|---|---|---|---|---|---|
| | Length | Time(s) | Length | Time(s) | Length | Time(s) |
| 2-opt | 1.871 | 0.21 | 2.340 | 1.60 | 2.732 | 4.02 |
| 2-opt+RTDL | 1.631 | 2.31 | 1.866 | 9.65 | 2.061 | 32.90 |
| 2-opt+RTDL+opt(D) | **1.581** | 4.16 | **1.832** | 18.53 | **2.037** | 50.65 |
| 3-opt | 1.161 | 28.05 | 1.234 | 346.36 | 1.304 | 1482.76 |
| 3-opt+RTDL | 1.156 | 29.00 | 1.210 | 329.49 | 1.282 | 1602.79 |
| 3-opt+RTDL+opt(D) | **1.144** | 22.03 | **1.203** | 310.40 | **1.278** | 1327.23 |

Table 5: Non-metric TSP. Improvements of 2-opt and 3-opt with RTDL barcodes.

Table 6: Average length of initial tours obtained via heatmap greedy decoding.

| Model | TSP-500 | TSP-1000 | TSP-10000 |
|---|---|---|---|
| DIFUSCO | 18.51 | 29.12 | 113.74 |
| ATT-GCN | 24.45 | 38.29 | 205.84 |
| DIMES | 29.17 | 46.61 | 331.73 |
| UTSP | 21.89 | 33.51 | – |
| SoftDist | 19.48 | 26.97 | 84.20 |

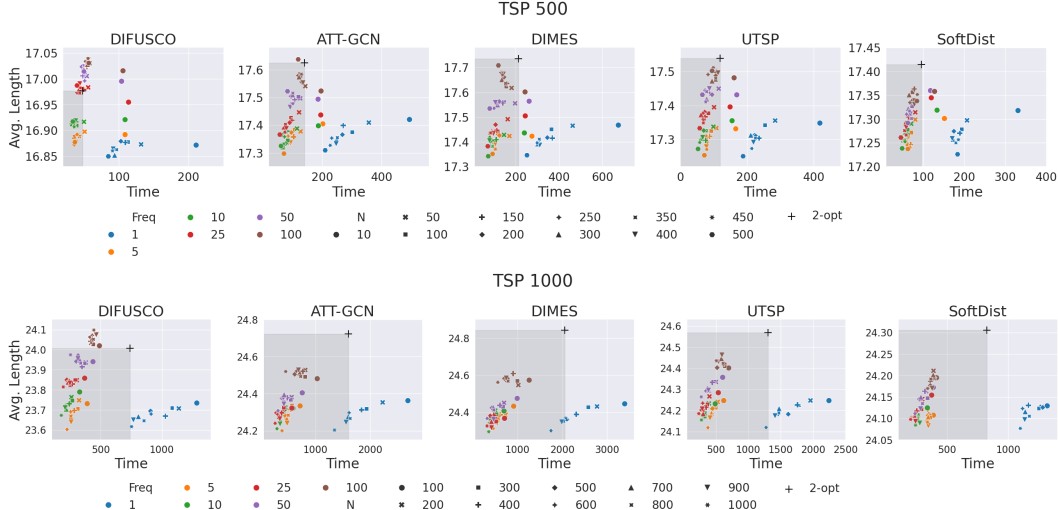

Figure 8: Sensitivity analysis for heatmap-guided 2-opt + RTDL. *Freq* refers to RTDL update frequency, *N* refers to number of vertices to optimize. Grey area indicates solutions better than 2-opt solution, i.e. shorter average length and faster computation.

Table 7: Comparison of failure cases for different TSP solvers on TSPLib instances. For each instance we report the percentage of runs where the constructed tour length exceeds $10\%$ above the global optimum ("Len. $> 10\%$ opt.") and the percentage of runs that failed to converge within the allocated time budget ("Time limit") in 20 seconds.

| | 2-opt | | 2-opt+RTDL-Full | | 2-opt+RTDL | |
| Name | GAP $> 10\%$ ↓ | Time limit (%) ↓ | GAP $> 10\%$ ↓ | Time limit (%) ↓ | GAP $> 10\%$ ↓ | Time limit (%) ↓ |
|---|---|---|---|---|---|---|
| ulysses16 | 0.0 | 0.0 | 0.0 | 0.0 | 0.0 | 0.0 |
| ulysses22 | 0.0 | 0.0 | 0.0 | 0.0 | 0.0 | 0.0 |
| att48 | 0.6 | 0.0 | 0.7 | 0.0 | **0.4** | 0.0 |
| eil51 | 0.7 | 0.0 | 0.9 | 0.0 | **0.5** | 0.0 |
| berlin52 | 27.1 | 0.0 | 23.5 | 0.0 | **20.7** | 0.0 |
| st70 | 5.6 | 4.0 | 5.9 | **0.0** | **4.0** | 0.0 |
| eil76 | 7.7 | 2.7 | 9.7 | **0.2** | **2.3** | 1.5 |
| pr76 | 1.6 | 0.0 | 2.7 | 0.0 | **1.2** | 0.0 |
| gr96 | 4.9 | 0.0 | 6.4 | 0.0 | **3.0** | 0.0 |
| kroC100 | 9.2 | 0.0 | 11.8 | 0.0 | **7.8** | 0.0 |
| rd100 | 12.25 | 0.0 | 19.0 | 0.0 | **9.0** | 0.0 |
| kroD100 | 6.75 | 0.0 | 6.75 | 0.0 | **3.75** | 0.0 |
| eil101 | 45.25 | 39.0 | 4.75 | **1.5** | **7.25** | 7.75 |
| lin105 | 12.25 | 0.5 | 19.0 | **0.25** | **12.0** | 0.5 |
| ch130 | 9.0 | 0.0 | 7.5 | 0.0 | **2.0** | 0.0 |
| ch150 | 11.2 | 0.0 | 19.2 | 0.0 | **5.0** | 0.0 |

limit"). The plain **2-opt** baseline shows the highest rate of such failures, while **2-opt+RTDL-Full** significantly reduces them. The **2-opt+RTDL** variant achieves the most robust performance overall, with consistently lower failure rates across different TSPLib instances.

# E    DQN ALGORITHM

In this section, we present the standard Deep Q-Learning (DQN) algorithm 3 applied to the TSP.

# F    ABLATION STUDY

In ablation study, we compare sorting edges by RTDL bars with sorting by distances in a descending order, see Table 8.

Figure 9: The distributions of calculation times and resulting tour lengths for different versions of 2-opt.

---

**Algorithm 3:** Deep Q-Learning (DQN) for TSP

---

Initialize replay buffer `D`;
Initialize Q-network with random weights $\theta$;
Initialize target network with weights $\theta_{\text{target}} \leftarrow \theta$;
Initialize environment with fixed city coordinates `env`;
Initialize `best_tour` and `best_length` $\leftarrow \infty$;

**for** $episode = 1$ **to** $M$ **do**
    $state \leftarrow$ `env.reset()`;
    **for** $t = 1$ **to** $T$ **do**
        With probability $\varepsilon$ select random action $a$, otherwise $a \leftarrow \arg\max_a Q(state, a; \theta)$;
        $state'$, `reward`, `done` $\leftarrow$ `env.step(a)`;
        Store $(state, a, \text{reward}, state', \text{done})$ in `D`;
        **if** $|D| \geq$ `batch_size` **then**
            Sample random minibatch of transitions from `D`;
            $y_j \leftarrow \text{reward}_j + \gamma \cdot (1 - \text{done}_j) \cdot \max_{a'} Q(state'_j, a'; \theta_{\text{target}})$;
            $L(\theta) \leftarrow \frac{1}{B} \sum_j \left(Q(state_j, a_j; \theta) - y_j\right)^2$;
            Update $\theta \leftarrow \theta - \alpha \cdot \nabla_\theta L(\theta)$;
        **end**
        **if** $t \bmod C = 0$ **then**
            $\theta_{\text{target}} \leftarrow \theta$;
        **end**
        **if** `done` **then**
            **if** *tour length* $<$ `best_length` **then**
                Update `best_tour` and `best_length`;
            **end**
            break;
        **end**
        $state \leftarrow state'$;
    **end**
**end**
**return** `best_tour`

---

| Method | TSP-100 | | TSP-200 | | TSP-300 | |
|---|---|---|---|---|---|---|
| | Length | Time(s) | Length | Time(s) | Length | Time(s) |
| 2-opt+dist. | 8.474 | 0.72 | 11.780 | 6.32 | 14.291 | 26.64 |
| 2-opt+alpha | 8.471 | 0.76 | 11.75 | 6.70 | 14.31 | 29.65 |
| 2-opt+RTDL | **8.226** | 2.84 | **11.356** | 18.08 | **13.798** | 62.30 |

Table 8: 2D Euclidean TSP. Ablation study of RTDL barcodes.

| | | | 2-opt | | 2-opt + RTDL | |
|---|---|---|---|---|---|---|
| Problem set | Optimal Length | | Length | Time | Length | Time |
| ATSP Uniform-50 | $1.55 \pm 0.18$ | | $2.08 \pm 0.26$ | 0.01 | $1.93 \pm 0.24$ | 0.14 |
| ATSP Uniform-100 | $1.57 \pm 0.13$ | | $2.11 \pm 0.2$ | 0.06 | $1.98 \pm 0.18$ | 0.63 |
| ATSP Uniform-200 | $1.56 \pm 0.11$ | | $2.16 \pm 0.15$ | 0.35 | $1.90 \pm 0.14$ | 4.01 |
| ATSP Uniform-500 | $1.57 \pm 0.06$ | | $2.22 \pm 0.10$ | 3.7 | $2.05 \pm 0.08$ | 38.4 |
| ATSP HCP-50 | 0.0 | | $4.27 \pm 1.84$ | 0.01 | $2.89 \pm 1.54$ | 0.1 |
| ATSP HCP-100 | 0.0 | | $4.01 \pm 1.85$ | 0.05 | $2.44 \pm 1.48$ | 0.43 |
| ATSP HCP-200 | 0.0 | | $3.37 \pm 1.48$ | 0.27 | $1.69 \pm 1.09$ | 2.7 |
| ATSP HCP-500 | 0.0 | | $3.4 \pm 1.18$ | 2.43 | $1.57 \pm 1.06$ | 9.44 |

Table 9: Average length and running time of 2-opt and 2-opt + RTDL algorithms applied to ATSP, HCP.

## G  LIMITATIONS

While our experiments focused on synthetic and benchmark TSPLIB instances, real-world route planning often involves additional constraints (time windows, capacities, dynamic updates) whose interaction with the topological reward scheme remains to be explored.

## H  NOTE ON DUPLICATE WEIGHTS

If some weights in a weighted graph are duplicate, MST can be non-unique. This can happen only if corresponding edges connect the same connected components in $G^{w<w(e)}$. The bijection between (s,t)-tour and MST is also non-unique in this case. However, the edge penalties $p(e) = w(e) - w(\psi(e))$ are still uniquely defined since $w(\psi(e))$ doesn't change.

## I  APPLICATION TO OTHER CO PROBLEMS

We provide additional evaluations for other combinatorial optimization problems – ATSP (Asymmetric TSP), HCP (Hamilton Cycle Problem, can be seen as a specific case of ATSP problem). We utilize the fact that non-symmetric TSP can be reduced to symmetric TSP by duplicating the number of vertices and transforming the cost matrix (see Jonker & Volgenant (1983)). For fair comparison, we apply both 2-opt and 2-opt + RTDL to ATSP, reduced to TSP. We utilize the ATSP and HCP problems from the benchmark (Ma et al., 2025). In the Table 9 we report average tour length and average time. The obtained results confirm that 2-opt + RTDL provides tours with smaller average length. We provide these results as a proof of concept that 2-opt + RTDL can be applied to other combinatorial optimization problems. Current implementation results in computational overhead compared to 2-opt and we leave its improvement to further work.

## J  THE USE OF LARGE LANGUAGE MODELS (LLMS)

We used LLMs to polish writing in the "introduction" and "abstract" sections.

