# OpenReview forum: "Edge-wise topological divergence gaps: guiding search in combinatorial optimization"
_ICLR.cc/2026/Conference — Submitted to ICLR 2026_

### Official Review · Reviewer_tLxQ · 2025-10-15

**Soundness:** 2
**Presentation:** 2
**Contribution:** 1
**Rating:** 2
**Confidence:** 4

**Summary:**

This paper proposes a topological framework for guiding search in combinatorial optimization, focusing on the Traveling Salesman Problem (TSP). The authors derive a canonical edge-wise decomposition between a TSP tour and its Minimum Spanning Tree (MST), expressing their cost gap as a sum of nonnegative “topological divergence” terms computed via the RTD-Lite barcode (a persistent homology construct). These divergence scores are then used to guide local search (2-opt/3-opt) and to shape reward signals in reinforcement learning (DQN-based) solvers.

**Strengths:**

The paper provides a fresh perspective on classical TSP analysis by introducing a canonical edge-wise decomposition that connects persistent homology with the MST–tour relationship. Unlike prior MST-based bounds or α-scores, this formulation produces a per-edge attribution of suboptimality derived from topological structure. This is a genuine conceptual contribution.

**Weaknesses:**

1.  Limited practical significance. As far as i understand: The proposed improvements are demonstrated only on simple 2-opt and 3-opt local search operators and a small DQN agent, Modern TSP solvers  employ far more complex and adaptive search strategies, making it unclear whether RTDL guidance would meaningfully improve real or industrial optimization pipelines, and also in data-driven fields, the search used in ATT-GCN or UTSP is much powerful than 2-opt, 3-opt.
2. MST-based reasoning is not new. The use of MSTs as lower bounds or structural references for TSP has a long history. The novelty here lies in the topological reinterpretation—the decomposition into edge-level bar lengths—not in the MST connection itself. This should be emphasized to avoid overstating originality.
3. Scalability and efficiency. RTDL computation introduces roughly 2–3× runtime overhead in some experiments.

**The most confusing aspect of the paper is the reinforcement learning component.**
The authors employ a basic Deep Q-Network (DQN) framework, which is now largely outdated for combinatorial optimization. Modern RL-based TSP solvers — such as POMO [Kwon et al., 2020], Attention Model [Kool et al., 2018] already achieve sub-1% optimality gaps on TSP-100 with inference times that are orders of magnitude faster. The paper does not compare against these established methods, making it unclear whether the proposed topological reward provides any meaningful improvement in a contemporary RL setting. Moreover, it is not well-motivated why the authors combine RL with simple 2-opt/3-opt post-processing rather than adopting the well-validated RL frameworks from prior work (e.g., POMO or the Attention Model), which already incorporate learned search or sampling strategies.
As a result, the RL results feel disconnected from current practice and do not convincingly demonstrate that the proposed topological feedback advances the state of the art in learning-based TSP solving.

**Questions:**

Could the RTDL-guided approach be integrated into deeper or variable-depth search frameworks?

How sensitive is the decomposition to small perturbations in edge weights (e.g., noisy or approximate distances)?

Can the topological divergence signal be used as a differentiable loss or regularizer in end-to-end neural solvers?

why we should use RL + 2/3 opt instead of using the RL style used in other works [Kwon et al., 2020] or [Kool 2018]?

---

> ### Author Response · Authors · 2025-11-26
> **Response to Reviewer tLxQ part 1/2**
>
> Thank you for your feedback. Below we respond to the questions one by one.
>
> **W1**. _Limited practical significance. The proposed improvements are demonstrated only on simple 2-opt and 3-opt local search operators and a small DQN agent, Modern TSP solvers employ far more complex and adaptive search strategies_
>
> **A**. The practical significance of our method is a fine-optimization of solutions from heatmap-based solvers DIFUSCO, ATT-GCN, DIMES, UTSP (Section 4.3) which are SOTA for neural TSP solvers. The fine-optimization is a common step after inferring a solution with greedy decoding (Sun & Yang, 2023). 2-opt+RTDL performs consistently better and faster than vanilla 2-opt.
> See also answer to Q4.
>
> **W2**. _MST-based reasoning is not new. The use of MSTs as lower bounds or structural references for TSP has a long history. The novelty here lies in the topological reinterpretation—the decomposition into edge-level bar lengths—not in the MST connection itself. This should be emphasized to avoid overstating originality._
>
> **A**. The use of MST as a lower bound for a full tour cost or structural reference (α-score) was studied before (Held & Karp, 1970), (Helsgaun, 2000). The prior work used MSTs as a lower bound for the full tour cost.
> By Theorem 1, we decompose the tour - MST gap and define a contribution of each tour’s edge (egde-level penalties). That is, our edge-level penalties are _tour dependent_. Existing edge scores (α-score) are defined for each edge of the graph without any relation to a tour and have been shown to not capture very well nontrivial global structure.
> Moreover, we prove that α-score is a particular case of RTDL barcode involving the graph consisting of one edge  (Section 3, Proposition 1).
>
> We are reformulating the relevant paragraph to make it more clear.
>
> **W3**. _Scalability and efficiency. RTDL computation introduces roughly 2–3× runtime overhead in some experiments._
>
> **A**. While for random instances 2-opt+RTDL is slower, for fine-optimization of tour from heatmaps 2-opt+RTDL is consistently faster than 2-opt. This happens because it takes less iterations to convergence, despite extra time taken to compute the RTDL barcode.
> Also, for random instances 3-opt+RTDL is consistently faster than 3-opt for the same reason.
>
> **W4.1**. _On reinforcement learning component of the paper_
>
> **A**. Our application of RTDL barcode to RL-based algorithms for TSP is a proof of concept. We empirically show that RTDL-based reward is useful. Experiments with more complicated deep RL models for TSP are in progress.
>
> **W4.2**. _it is not well-motivated why the authors combine RL with simple 2-opt/3-opt post-processing rather than adopting the well-validated RL frameworks from prior work (e.g., POMO or the Attention Model)_
>
> **A**. See answer to Q4.
>
>
>
> **Q1**. _Could the RTDL-guided approach be integrated into deeper or variable-depth search frameworks?_
>
> **A**. We integrated the RTDL-guided approach into the Lin-Kernighan algorithm  (https://github.com/kikocastroneto/lk_heuristic). The results are below (LK vs. LK+rtdl). We didn’t observe systematic improvement in performance. However, 3-opt+rtdl consistently outperforms LK in terms of time and tour length.
>
> | Method | Avg. tour. len. | Time. sec | Avg. tour. len. | Time. sec | Avg. tour. len. | Time. sec |
> |-|-|-|-|-|-|-|
> | | TSP-100 | | TSP-200 | | TSP-300 | |
> | 2opt+rtdl | 8,170 | 277,743 | 11,303 | 1681,830 | 13,654 | 5703,617 |
> | 3opt+rtdl | 8,011 | 1143,246 | 11,126 | 10461,307 | 13,546 | 45666,931 |
> | LK | 8.022 | 1749.791 | 11.176 | 11106.476  | 13.672 | 35095.393 |
> | LK+rtdl | 7.984 | 1828.717 | 11.228 | 11212.972 | 13.741 | 34623.934 |
>
> **Q2**. _How sensitive is the decomposition to small perturbations in edge weights (e.g., noisy or approximate distances)?_
>
> **A**. The tour-MST gap (equation (1)) is equal to the difference between sum of weights in (s,t)-tour and MST: $L_{(s,t)-tour}-L_{mst}$.
> Obviously, they change continuously w.r.t. small perturbation of edge weights.
> The one-to-one correspondence $\phi(\cdot)$ between edges of s,t-tour and MST remain unchanged if perturbations don't change ordering of edges according to weights.
>
> **Q3**. _Can the topological divergence signal be used as a differentiable loss or regularizer in end-to-end neural solvers?_
>
> **A**. RTDL is differentiable (Tulchinskii et al., 2025), its applications as a loss or a regularizer in end-to-end neural solvers is an interesting topic for further research.

---

> > ### Author Response · Authors · 2025-11-26
> > **Response to Reviewer tLxQ part 2/2**
> >
> > **Q4**. _Why we should use RL + 2/3 opt instead of using the RL style used in other works [Kwon et al., 2020] or [Kool 2018]?_
> >
> > **A**. SOTA methods like DIFUSCO (Sun & Yang, 2023) outperform AM (Kool 2018) and POMO (Kwon et al., 2020). However, a fine-optimization with 2-opt is applied in many recent methods: DIFUSCO (Sun & Yang, 2023), MatDIFFNet (Pan, 2025), GenSCO (Li, 2025), T2T (Li, 2023), EAN (Deudon, 2018).
> >
> > Deudon, M., Cournut, P., Lacoste, A., Adulyasak, Y., & Rousseau, L. M. (2018). Learning heuristics for the tsp by policy gradient. In International conference on the integration of constraint programming, artificial intelligence, and operations research (pp. 170-181). Cham: Springer International Publishing.
> >
> > Pan, W., Xiong, H., Ma, J., Zhao, W., Li, Y., & Yan, J. (2025). UniCO: On unified combinatorial optimization via problem reduction to matrix-encoded general TSP. In The Thirteenth International Conference on Learning Representations.
> >
> > Li, Y., Chen, L., Wang, H., Wang, R., & Yan, J. (2025). Generation as search operator for test-time scaling of diffusion-based combinatorial optimization. In The Thirty-ninth Annual Conference on Neural Information Processing Systems.
> >
> > Li, Y., Guo, J., Wang, R., & Yan, J. (2023). T2t: From distribution learning in training to gradient search in testing for combinatorial optimization. Advances in Neural Information Processing Systems, 36, 50020-50040.

---

### Official Review · Reviewer_P95f · 2025-10-27

**Soundness:** 3
**Presentation:** 2
**Contribution:** 2
**Rating:** 4
**Confidence:** 3

**Summary:**

This paper proposes a topological feedback mechanism for improving combinatorial optimization, specifically the Travelling Salesman Problem (TSP). The core idea is to measure the divergence between a candidate tour and the instance’s minimum spanning tree (MST) using edge-wise topological divergence gaps derived from the RTD-Lite barcode. Extensive experiments on TSPLIB, random Euclidean TSPs, and heatmap-based neural TSP solvers demonstrate improved performance—shorter tours, faster convergence, and greater stability—compared to standard 2-opt, 3-opt, and RL baselines.

**Strengths:**

1. The decomposition theorem provides a mathematically grounded link between graph topology and optimization performance.
2. Edge-wise topological divergence gives interpretable insight into why certain edges are suboptimal.

**Weaknesses:**

1. Although the authors claim broader applicability to combinatorial optimization, all experiments focus solely on TSP.
2. Although an ablation is included, the effect of RTDL frequency, edge selection granularity, and reward scaling could be analyzed more systematically.

**Questions:**

1. Does the bijection between tour and MST edges always hold uniquely? What happens for degenerate weight cases?
2. Could RTDL feedback be integrated during neural tour construction rather than post-hoc?

---

> ### Author Response · Authors · 2025-11-27
>
> Thank you for your feedback. Below we respond to the questions one by one.
>
> **Q1**. _Does the bijection between tour and MST edges always hold uniquely? What happens for degenerate weight cases?_
>
> **A**. If some weights in a weighted graph are duplicate, MST can be non-unique. This can happen only if corresponding edges connect the same connected components. The bijection between (s,t)-tour and MST is also non-unique in this case.
> However, the penalties $p(e) = w(e) - w(\psi(e))$ are uniquely defined since  $w(\psi(e))$ doesn’t change. We are updating our manuscript.
>
> **Q2**. _Could RTDL feedback be integrated during neural tour construction rather than post-hoc?_
>
> **A**. Yes, we use a topological-derived reward $r_{\text{RTDL}}(e) = w(e) - w(\psi(e))$ in experiments with neural tour construction in Section 4.4.
>
> **W1**. _Although the authors claim broader applicability to combinatorial optimization, all experiments focus solely on TSP._
>
> **A**. We provide additional evaluations for other combinatorial optimization problems – ATSP (Asymmetric TSP), HCP (Hamilton Cycle Problem, can be seen as a specific case of ATSP problem). We utilize the fact that non-symmetric TSP can be reduced to symmetric TSP by duplicating the number of vertices and transforming the cost matrix (see [1]). For fair comparison, we apply both 2-opt and 2-opt + RTDL to ATSP, reduced to TSP. We utilize the ATSP and HCP problems from the benchmark [2]. In the table below we report average tour length and average time. The obtained results confirm that 2-opt + RTDL provides tours with smaller average length. We provide these results as a proof of concept that 2-opt + RTDL can be applied to other combinatorial optimization problems. Current implementation results in computational overhead compared to 2-opt and we leave its improvement to further work.
>
> | | Opt. Len. | 2-opt | | 2-opt + RTDL | |
> |-|-|-|-|-|-|
> | | | Len. | Time | Len. | Time |
> | ATSP Uniform-50| 1,55 ± 0,18 | 2,08 ± 0,26 | 0,01 | 1,93 ± 0,24 | 0,14 |
> | ATSP Uniform-100| 1,57 ± 0,13 | 2,11 ± 0,2 | 0,06 | 1,98 ± 0,18 | 0,63 |
> | ATSP Uniform-200| 1,56 ± 0,11 | 2,16 ± 0,15 | 0,35 | 1,90 ± 0,14 | 4,01|
> | ATSP Uniform-500| 1,57 ± 0,06 | 2,22 ± 0,10 | 3,7 | 2,05 ± 0,08 | 38,4 |
> | ATSP HCP-50 | 0,0 | 4,27 ± 1,84 | 0,01 | 2,89 ± 1,54 | 0,1|
> | ATSP HCP-100| 0,0 | 4,01 ± 1,85 | 0,05 | 2,44 ± 1,48 | 0,43 |
> | ATSP HCP-200| 0,0 | 3,37 ± 1,48 | 0,27 | 1,69 ± 1,09 | 2,7 |
> | ATSP HCP-500| 0,0 | 3,4 ± 1,18 | 2,43 | 1,57 ± 1,06 | 9,44 |
>
> [1] Jonker, Roy; Volgenant, Ton. Transforming asymmetric into symmetric traveling salesman problems. 1983.
> [2] Jiale Ma, Wenzheng Pan, Yang Li, Junchi Yan. ML4CO-Bench-101: Benchmark Machine Learning for Classic Combinatorial Problems on Graphs. NeurIPS 2025.
>
> **W2**. _Although an ablation is included, the effect of RTDL frequency, edge selection granularity, and reward scaling could be analyzed more systematically._
>
> **A**. We can note that 2-opt+RTDL with Freq=1 (i.e., update on each iteration) typically leads to higher computation time, for all values of granularity. In the majority of cases, the average tour length for Freq=1 can be matched or even improved with Freq=5, 10 with appropriate granularity with sufficiently smaller computation time. Overall, for Freq>1, more frequent updates lead to smaller average tour length.
>
> In general, the larger the granularity, the slower is computation, although this effect is less pronounced for higher frequencies. Up to some threshold frequency, larger granularity leads to higher average tour length. Above the threshold frequency, larger granularity leads to decreased average tour length. This threshold frequency depends on model and problem size.

---

### Official Review · Reviewer_GRro · 2025-10-29

**Soundness:** 3
**Presentation:** 3
**Contribution:** 3
**Rating:** 4
**Confidence:** 5

**Summary:**

This submission deals with different heuristics for approximating optimal TSP tours. The novel contribution is to assign to each edge in a given (non-optimal) tour a value that intuitively measures the contribution of the edge to the non-optimality of the tour. Then these values are used to guide the local search heuristics 2-Opt and 3-Opt and reinforcement learning. The values are computed via RTD-barcodes, a topological method from existing literature to compare weighted graphs.

The focus of the results is on experimental evaluation of the heuristics and not on theoretical considerations. First, the authors consider 2-Opt and 3-Opt and use the topological values to prioritize which local improvements are performed (i.e., first edges with large values are tried to be swapped out). They show on several instances that these guided local search algorithms are superior to arbitrary local search algorithms that simply perform the first local improvement found. In particular, the quality of the resulting solutions improves slightly both for 2-Opt and 3-Opt and the running time improves for 3-Opt. Then the authors consider large-scale instances, where the initial solutions are computed via heatmap approaches. Also for these, the topologically guided version of 2-Opt is slightly better in terms of the resulting tour length and better in terms of running time. Finally, the authors also consider Q-Learning. They demonstrate that incorporating the topological values into the learning process leads to a speed up.

**Strengths:**

Local search and other heuristics for the TSP are an interesting area of research that has been an active field for decades. Hence, the line of research is interesting and well-motivated. Using a topological measure to speed up these heuristics is interesting. It is also interesting that this measure can be used in different heuristics (local search and Q-learning).

**Weaknesses:**

Please see the text in the field "Questions" below for a detailed discussion of the weaknesses. In summary, the quality of the writing could be improved and more experiments are necessary in my opinion.

**Questions:**

- The proof of Theorem 1 is fairly confusing and could be written more precisely. In particular, it could be elaborated more why the constructed mapping is injective.
- I find the paragraph in lines 151 to 161 quite unclear: RTDL barcodes are briefly mentioned and I get that for any two undirected unweighted graphs with a bijection of the nodes such a barcode can be obtained via Algorithm 1 from (Tulchinskii et al., 2025). Then the barcode for the current tour T_tour and the complete graph G is computed. But how is this information used? Further below, the penalties are defined as p(e)=w(e)-w(psi(e)), which is related to Theorem 1 but, as far as I understand the notation, unrelated to the barcode computed before. Which values are assigned to the edges? If only the values p(e) are used then I do not understand the role of the barcodes at all because these values depend only on the procedure in Theorem 1.
- Throughout the paper it is said that random instances are generated but it is not specified what distributions are used.
- Is there any intuition why setting p(e_max) to the minimum of the positive p(e) values leads empirically to the best results? On the first glance, this is not the most intuitive choice.
- Some more experimental comparisons would be interesting/necessary. In particular, the following:
  1) How do the proposed algorithms (like 2-Opt+RTDL) perform compared with state-of-the-art heuristics like Lin-Kernighan?
  2) Is there already literature on the effect of which local improvements are chosen in 2-Opt or 3-Opt? At least variants that find the first improvement have been compared with variants that find the largest improvement (see references below).
  3) The alpha-score of LKH-3 is mentioned in the submission. Couldn't one use this score directly to guide the local search instead of the values used in this submission? How does this compare to the method proposed in the submission?

Pierre Hansen, Nenad Mladenović,
First vs. best improvement: An empirical study,
Discrete Applied Mathematics,
Volume 154, Issue 5,
2006,

Daniel Aloise, Robin Moine, Celso C. Ribeiro, Jonathan Jalbert,
First-improvement or best-improvement? An in-depth local search computational study to elucidate a dominance claim,
European Journal of Operational Research,
Volume 326, Issue 3,
2025,

Typos:
Line 60: deliver -> delivers

Line 139: denotes -> denote

Line 203: if -> is

Line 237: either leave out "a" or tours -> tour

Line 241: note -> noted

Line 432: signal -> signals

---

> ### Author Response · Authors · 2025-11-25
> **Response to Reviewer GRro**
>
> Thank you for your feedback. Below we respond to the questions one by one.
>
> **Q1**. _The proof of Theorem 1 is fairly confusing and could be written more precisely. In particular, it could be elaborated more why the constructed mapping is injective._
>
> **A**.  The injectivity follows for example from the barcode interpretation of the construction from the proof, as the correspondence between birth and death segments in persistence pairs is bijective. We are adding details for the Theorem 1 proof in the revision, see Appendix B.
>
> **Q2**. _On clarity of lines 151 to 161, RTDL barcodes and values p(e)._
>
> **A**. Thank you for the remark. RTDL-Barcode is a sequence of intervals:
>
>  $$
> \text{RTDL-Barcode}(T_{\text{tour}}, G) = \\{ (w(e), w(\phi(e)))  |   e\in E(T_{mst}) \\}
> $$
> and the gap
> $$
>     L_{(s,t)-tour}-L_{mst}=\sum_{e\in E(T_{mst})} w(\phi(e))-w(e)
> $$
> from Theorem 1 is equal to the sum of lengths of bars in $\text{RTDL-Barcode}(T_{\text{tour}}, G)$.
>
> We use length of each interval in $\text{RTDL-Barcode}(T_{\text{tour}}, G)$ as edge’s badness: $p(e) = w(\phi(e)) - w(e) $
>
> We are including these clarifications in Section 3.
>
> **Q3**. _Throughout the paper it is said that random instances are generated but it is not specified what distributions are used._
>
> **A**. For Euclidean TSP, coordinates of cities are sampled from a unit square. For non-metric TSP, weights are sampled uniformly from (0, 1). We are adding this information to the paper.
>
> **Q4**. _Is there any intuition why setting p(e_max) to the minimum of the positive p(e) values leads empirically to the best results? On the first glance, this is not the most intuitive choice._
>
> **A**.  This equation was selected empirically in ablation study. $p(e_{max})$ defines the order in which $e_{max}$ is considered in 2-opt/3-opt. Edges with $p(e)=0$ belong to MST.
> We analyzed the following variants:
> * $p(e_{max}) = w(e_{max})$; $e_{max}$ is considered before all the other edges;
> * $p(e_{max}) = 0$; $e_{max}$ is considered after all edges with $p(e)>0$, the order between edges with $p(e) = 0$ is not specified;
> * $p(e_{max}) $ = the minimum of the positive $p(e)$ values (not including $e_{max}$);  $e_{max}$ is considered after all edges with $p(e)>0$, and before edges with $p(e) = 0$.
>
> **Q5.1**. _How do the proposed algorithms (like 2-Opt+RTDL) perform compared with state-of-the-art heuristics like Lin-Kernighan?_
>
> **A**. The comparison with Lin-Kernighan is below:
>
> | Method | Avg. tour. len. | Time, sec |Avg. tour. len. | Time, sec |Avg. tour. len. | Time, sec |
> | - | - | - | - | - | - | - |
> | |TSP-100 | |TSP-200 | | TSP-300 | |
> | 2opt+rtdl | 8,170 | 277,743 | 11,303 | 1681,830 |13,654 | 5703,617 |
> | 3opt+rtdl | 8,011 | 1143,246 | 11,126 | 10461,307 | 13,546 | 45666,931 |
> | LK | 8,022 | 1749,791 | 11,176 | 11106,476 | 13,672 | 35095,393 |
>
> We used a Python implementation https://github.com/kikocastroneto/lk_heuristic. 2-opt/3-opt algorithms employed optimized distance matrices (see Section 4.2), results are averaged over 100 trials.
>
> **Q5.2**. _Is there already literature on the effect of which local improvements are chosen in 2-Opt or 3-Opt? At least variants that find the first improvement have been compared with variants that find the largest improvement._
>
> **A**. Thank you for the remark. We are revising the related work section to include a dedicated discussion on the “first vs. best improvement” and adding (Hansen et al., 2006), (Aloise et al., 2025) to References.
> Our implementations of 2-opt/3-opt with and without RTDL follow the “first improvement” rule. The RTDL “badness” score affects order of edges in 2-opt/3-opt.
> Edges having higher RTDL score (and lower probability of belonging to an optimal tour, see Fig. 2) are picked earlier by the algorithm.
>
> **Q5.3**. _Couldn't one use the alpha-score from LKH-3 directly to guide the local search instead of the values used in this submission?_
>
> **A**. We modified 2-opt to rank edges by alpha-score from LKH-3. Since both RTDL penalty and alpha-score represent edge’s “badness”, rest of the Algorirth remained unchanged. Results are below. 2-opt+RTDL finds a better tour, but it is slower because computation of RTDL barcodes takes extra time.
>
> | Method | Avg. Len | Time, sec | Avg. Len | Time, sec | Avg. Len | Time, sec |
> | - | - | - | - | - | - | - |
> | |TSP-100 | |TSP-200 | | TSP-300 | |
> | 2-opt+RTDL | 8.226 | 2.84 | 11.35 | 18.08 | 13.79 | 62.30 |
> | 2-opt+alpha |  8.471 | 0.76 | 11.75 | 6.70  | 14.31 | 29.65 |

---

### Official Review · Reviewer_cKwN · 2025-11-05

**Soundness:** 3
**Presentation:** 3
**Contribution:** 3
**Rating:** 4
**Confidence:** 2

**Summary:**

The paper defines an edge-wise topological divergence score that measures how much each tour edge deviates from the structure of the minimum spanning tree (MST). By prioritizing edges with higher divergence during local search and using the same scores as a shaping signal in reinforcement learning, the method achieves faster convergence and produces shorter tours across classical, neural, and RL-based TSP solvers.

**Strengths:**

- The proposed method can be integrated with numerical and learning based solvers
- Experimental results show that the proposed method provides faster convergence and shorter tours.

**Weaknesses:**

- The core optimization idea is not fundamentally new. The paper’s edge badness score closely parallels the α-score in LKH-3, which already uses MST/1-tree structure to rank edges for removal.
- The paper doesn't compare to state of the art TSP solvers like Concorde, LKH.
- The RL experiments are limited to relatively small problem sizes (100 nodes). It is unclear whether the RTDL-shaped reward remains stable and beneficial for larger tours.

**Questions:**

- What conceptual advantage does the topological interpretation provide over the classical α-score heuristic beyond reinterpretation?

---

> ### Author Response · Authors · 2025-11-24
> **Response to Reviewer cKwN**
>
> Thank you for your feedback. Below we respond to the questions one by one.
>
> **W1**. _The core optimization idea is not fundamentally new. The paper’s edge badness score closely parallels the α-score in LKH-3, which already uses MST/1-tree structure to rank edges for removal._
>
> **Q1**. _What conceptual advantage does the topological interpretation provide over the classical α-score heuristic beyond reinterpretation?_
>
> **A**. The main difference between our edge penalty (badness) and the α-score is that our penalty is tour-dependent. The α-score in LKH-3 is defined for each edge of the graph without any relation to the tour. Also, contrary to the α-score, our penalties decompose the standard tour length-MST gap into the sum of non-negative gaps corresponding to the tour edges. We prove that α-score is a particular case of RTDL barcode involving the graph consisting of one edge (Section 3, Proposition 1). We are updating our manuscript to make it more clear.
>
> **W2**. _The paper doesn't compare to state of the art TSP solvers like Concorde, LKH._
>
> **A**. We provide a comparison of the best heatmap-based method with Concorde and LKH-3 below.
>
> | | TSP-500 | | TSP-1000 | | TSP-10000 | |
> |-|-|-|-|-|-|-|
> | | Avg. Length | Avg. Time (s) |  Avg. Length | Avg. Time (s) | Avg. Length |  Avg. Time (s) |
> | Concorde | 16.55 | 18.73 | 23.12 | 201.01 |  N/A | N/A |
> | LKH-3 | 16.55 | 41.75 | 23.12 | 90.94 | 71.77 | 2698.38 |
> | DIFUSCO+2-opt | 16.98 | 0.38 | 24.01 | 5.78 | 75.87 | 7027.63 |
> | DIFUSCO+2-opt + RTDL | 16.88 | 0.23 | 23.60 | 1.76 | 74.28 | 865.5 |
>
> We conclude that DIFUSCO is slightly worse than LKH-3 and Concorde which reflects the current performance of this family of methods. At the same time, DIFUSCO+2-opt + RTDL is faster than Concorde and LKH-3 and finds shorter tours than DIFUSCO+2-opt.
>
>
> **W3**. _The RL experiments are limited to relatively small problem sizes (100 nodes). It is unclear whether the RTDL-shaped reward remains stable and beneficial for larger tours._
>
> **A**. The RTDL-shaped reward remains stable beyond 100 nodes. We carried out experiments with 200 and 500 nodes. Despite this, our current DQN-based experiments are only a proof of concept, since this architecture cannot scale to large TSP instances like modern solvers. The limitation is in the model, not in the reward design.

---

### Author Response · Authors · 2025-11-28
**General response**

We sincerely thank the Reviewers for their insightful comments and the opportunity to strengthen our work.
In this global response, we address the most frequent topics and highlight changes in the manuscript.
For ease of reading, changes are highlighted in yellow.
We are happy to address any additional questions during the discussion phase.

1) **On the conceptual advantage of RTDL barcodes over the classical α-score**.

* The main difference between our edge penalty (badness) and the α-score is that our penalty is tour-dependent. The α-score in LKH-3 is defined for each edge of the graph without any relation to the tour. Also, contrary to the α-score, our penalties decompose the standard tour length-MST gap into the sum of non-negative gaps corresponding to the tour edges. We prove that α-score is a particular case of RTDL barcode involving the graph consisting of one edge (Section 3, Proposition 1). We have updated Section 3.

* A comparison with the modification of 2-opt employing α-score was added to Table 8 in Appendix.
2-opt+RTDL finds better tours than 2-opt+alpha, but it is slower because computation of RTDL barcodes takes extra time.

2) **On the comparison with Concorde, LKH-3**.

We have added a comparison of heatmap-based methods with Concorde and LKH-3, see Table 2.
The top-performing method, DIFUSCO, is slightly worse than LKH-3 and Concorde which reflects the current performance of this family of methods. At the same time, DIFUSCO+2-opt + RTDL is faster than Concorde and LKH-3 and finds shorter tours than DIFUSCO+2-opt.

3) **On broader applicability to combinatorial optimization, besides TSP.**

We have added comparisons of 2-opt vs. 2-opt+RTDL applied to ATSP (Asymmetric TSP), HCP (Hamilton Cycle Problem), see Appendix I.
The obtained results confirm that 2-opt + RTDL provides tours with smaller average length. We provide these results as a proof of concept that 2-opt + RTDL can be applied to other combinatorial optimization problems. Current implementation results in computational overhead compared to 2-opt and we leave its improvement to further work.

4) **Limited practical significance. The proposed improvements are demonstrated only on simple 2-opt and 3-opt local search operators and a small DQN agent. Modern TSP solvers employ far more complex and adaptive search strategies**.

The practical significance of our method is a fine-optimization of solutions from heatmap-based solvers like DIFUSCO (Section 4.3) which are SOTA for neural TSP solvers. The fine-optimization is a common step after inferring a solution with greedy decoding DIFUSCO (Sun & Yang, 2023), MatDIFFNet (Pan, 2025), GenSCO (Li, 2025), T2T (Li, 2023), EAN (Deudon, 2018). 2-opt+RTDL performs consistently better and faster than vanilla 2-opt.

5) **On clarity of lines 151 to 161, RTDL barcodes and values p(e).**

We have updated Section 3.

6) **Throughout the paper it is said that random instances are generated but it is not specified what distributions are used.**

We have added a clarification to Section 4.2.

7) **On comparison of 2-opt/3-opt with Lin-Kernighan.**

We have added a comparison to Table 1. 3-opt+rtdl consistently outperforms LK in terms of time and tour length.

8) **On first improvement vs. best improvement.**

We have added references (Hansen, 2006), (Aloise, 2025) to the Related Work section.

9) **On the bijection between tour and MST edges, what happens for degenerate weight cases.**

We have added details on bijection in Theorem 1 in Appendix B, and a note on degenerate weights in Appendix H.

10) **Several typos were fixed.**

-----
Best regards, the Authors.

---

### Meta-Review · Area_Chair_D6Bn · 2026-01-04

**Summary:**

1. The generalization of the RL-based augmented approach is weak, which can only handle 500 points.
2. When the size is small, the performance is weaker than the SOTA solver, including Concorde or LKH-3

**Reviewer Concerns:**

Addressed problem:
1. The technical novelty compared to $\alpha$-score
2. The criticism of only working on the TSP problem

Remained problem:
1. The significance of the performance improvement by the proposed algorithm compared to the SOTA approaches.

**Reviewer Scores:**

No change

---

### Decision · Program_Chairs · 2026-01-26

Reject